# Association of Plasma Total Cysteine and Anthropometric Status in 6–30 Months Old Indian Children

**DOI:** 10.3390/nu12103146

**Published:** 2020-10-15

**Authors:** Catherine Schwinger, Ranadip Chowdhury, Shakun Sharma, Nita Bhandari, Sunita Taneja, Per M. Ueland, Tor A. Strand

**Affiliations:** 1Centre for Intervention Science in Maternal and Child Health, Centre for International Health, Department of Global Public Health and Primary Care, University of Bergen, Catherine Schwinger, Årstadveien 21, 5009 Bergen, Norway; ranadip.chowdhury@sas.org.in (R.C.); nita.bhandari@sas.org.in (N.B.); Tor.Strand@uib.no (T.A.S.); 2Society for Applied Studies, New Delhi 110016, India; sunita.taneja@sas.org.in; 3Department of Child Health, Institute of Medicine, Tribuhvan University, Kathmandu 44613, Nepal; shakunsharma1@gmail.com; 4Department of Clinical Science, University of Bergen,5020 Bergen, Norway; per.ueland@ikb.uib.no; 5Department of Research, Innlandet Hospital Trust, 2618 Lillehammer, Norway

**Keywords:** wasting, stunting, underweight, child growth, low- and middle-income countries (LMIC), malnutrition, amino acid, metabolism, pre-school children, under-5 children

## Abstract

High-quality protein has been associated with child growth; however, the role of the amino acid cysteine remains unclear. The aim was to measure the extent to which plasma total cysteine (tCys) concentration is associated with anthropometric status in children aged 6–30 months living in New Delhi, India. The study was a prospective cohort study including 2102 children. We calculated Z-scores for height-for-age (HAZ), weight-for-height (WHZ), or weight-for-age (WAZ) according to the WHO Child Growth Standards. We used multiple regression models to estimate the association between tCys and the anthropometric indices. A high proportion of the children were categorized as malnourished at enrolment; 41% were stunted (HAZ ≤ −2), 19% were wasted (WHZ ≤ −2) and 42% underweight (WAZ ≤ −2). Plasma total cysteine (tCys) was significantly associated with HAZ, WHZ and WAZ after adjusting for relevant confounders (*p* < 0.001). Low tCys (≤25th percentile) was associated with a decrease of 0.28 Z-scores for HAZ, 0.10 Z-scores for WHZ, and 0.21 Z-scores for WAZ compared to being >25th percentile. In young Indian children from low-to-middle socioeconomic neighborhoods, a low plasma total cysteine concentration was associated with an increased risk of poor anthropometric status.

## 1. Introduction

Child undernutrition is a significant public health problem in developing countries. It is estimated that globally, 150.8 million children under the age of 5 are stunted and 50.5 million are wasted [1]. These forms of malnutrition are associated with adverse short- and long-term consequences for growth and development [2,3,4] as well as for survival [5,6].

Young children grow rapidly and have additional nutritional demands to enhance optimal growth and development [7]. These demands are increased by environmental influences, such as a high burden of clinical and sub-clinical infections [8,9,10]. Intake of high-quality protein has been shown to promote childhood growth [9,10,11,12,13] with a suggested pathway via the production of insulin-like growth factor I (IGF-I). Additionally, a sensing pathway, the mechanistic target of rapamycin (mTOR), has been described, which regulates energy balance and suppresses cellular growth in the case of amino acid deficiency [11,14]. In a study among Malawian children, stunted children had lower serum concentrations of essential, conditionally essential, and non-essential amino acids compared to non-stunted children [11]. However, there is no evidence that the conditionally essential amino acid cysteine is associated with linear growth in children. Cysteine is an important component in the structure and function of proteins and enzymes and is the rate-limiting precursor of the antioxidant glutathione, which is involved in the prevention of cell damage and functioning of the immune system [15].

Animal studies have shown that cysteine supplementation in mice increased growth plate thickness through upregulation of insulin-like growth factor-I (IGF-1), and led to improvements in different bone parameters [16]. Thus, there is a biological plausibility that cysteine supplementation can increase IGF-1 levels in children, which can improve linear growth. If cysteine supplementation would show to improve linear growth in children as well, this could contribute to alleviating the high burden stunting, especially in countries such as India. However, as a first step, the association between cysteine and linear growth should be elucidated in a child population.

Further, high cysteine levels have repeatedly been associated with adult obesity based on evidence from cellular, animal, and epidemiological studies [17,18,19,20], but here is only scarce evidence on the associations of low levels of cysteine and undernutrition during childhood with a few studies focusing on edematous children [21,22,23], or on pregnancy outcomes such as preterm birth and low birth weight [24,25].

In this secondary analysis, we aimed to assess to what extend plasma total cysteine (tCys) concentration is associated with poor anthropometric status in children aged 6–30 months living in low-to-middle socioeconomic neighborhoods of New Delhi, India.

## 2. Materials and Methods

### 2.1. Original Study

We used data from a randomized controlled trial (NCT00272116 at www.clinicaltrials.gov), which took place from 1998 until 2000 in New Delhi, India. It evaluated the effect of daily zinc supplementation on the incidence of acute lower respiratory tract infections and pneumonia [26]. Details are described elsewhere [26]. In brief, the trial enrolled 2482 children aged 6–30 months, living in the community of Dakshinpuri that had around 75,000 inhabitants at the time of the study. All children aged 6–30 months were identified through a survey. Exclusion criteria were no informed consent, plans to move out of the study area within the next four months, the need for urgent admission to the hospital, and a recent massive dose of vitamin A (100,000 IU for infants; 200,000 for older children). All analyses in the current study are restricted to the group of 2102 children whose tCys concentration and anthropometry measurements were available.

Ethical clearance for the trial was given by the All India Institute of Medical Sciences Ethics Committee. In the main trial, written informed consent was obtained from the caretaker; permission was also sought to store the children’s blood specimen for use in future research. All parents consented to this. All activities were performed in accordance with the Declaration of Helsinki.

### 2.2. Measurements

All measurement procedures were described in a manual, and standardization exercises were done every three months. Supervisors monitored all field workers’ activities and independently checked data collection (1% of visits).

Anthropometric measurements and non-fasting venous blood samples were taken at enrollment and the end of the study, i.e., exactly four months after enrollment. Weight was measured with electronic scales (seca), reading to the nearest 100 g. Length (or height for children ≥ 24 months) was measured with locally manufactured length boards with a precision of 0.1 cm. Heavy clothing and nappies were removed before measurements. Blood samples (≈5 mL) were collected between 900 h and 1200 h in heparinized polypropylene tubes (Sarstedt) by a study physician. The samples were centrifuged, and plasma was stored immediately in polypropylene vials (Eppendorf) at −20 °C for subsequent analysis. A microbiological assay with a chloramphenicol-resistant strain of *Lactobacillus casei* was used to measure plasma cobalamin concentrations [27], while a colistin sulfate–resistant strain of *Lactobacillus leichmannii* was used for plasma folate [28]. Measurements were done by a robotic workstation using a microtiter plate format. Plasma methylmalonic acid (MMA), plasma total homocysteine (tHcy) and plasma total cysteine (tCys) concentrations were measured with a modified gas chromatography–mass spectrometry method based on ethylchloroformate derivatization [29]. The between-day coefficient of variation is reported as <5% for all assays except for tCys, where it is ~10% [27,28,29]. For cobalamin, we calculated a composite measure status (3cB12) according to the formula presented by Fedosov, et al. [30]. The composite score we used combines three biomarkers of vitamin B_12_ (total B_12_, MMA, tHcy) and corrects for age and folate status. This is suggested to increase the precision of the diagnosis of vitamin B_12_ deficiency [30]. We included 3cB12 because cobalamin is an important factor in the transsulfuration pathway from methionine to cysteine [31].

Upon enrolment, the study physician also interviewed the caretaker including questions on the birthdate of the child, breastfeeding practices, years of education of the parents, family size and household income.

### 2.3. Data management and Analysis

Statistical analyses were done with STATA 15.0 statistical software (StataCorp, College Station, TX) and R version 3.5.1 (R Foundation for Statistical Computing, Vienna, Austria).

We calculated Z-scores for height-for-age (HAZ), weight-for-height (WHZ) and weight-for-age (WAZ) according to the WHO Child Growth Standards [32]. Stunting, wasting and underweight are defined as Z-scores ≤ −2 for height-for-age, weight-for-height, and weight-for-age, respectively.

Baseline characteristics are reported as mean (SD), median (IQR) or proportion, as appropriate. We dichotomized values for plasma cysteine at the 25th percentile because established reference values for plasma cysteine concentrations do not exist. Differences in means, medians or proportions between children with a tCys concentration above the 25th percentile compared to those on the 25th percentile or below were tested with an unpaired *t*-test, a Wilcoxon rank-sum test or a Chi2-Test, respectively.

We built generalized estimating equation (GEE) models with anthropometric indices as outcome and tCys concentration at enrollment as exposure. This approach uses growth data from both time points (enrollment and 4 months later) and accounts for interdependence between multiple measurements in the same child [33]. For continuous outcomes (HAZ, WHZ, WAZ), we used GEE models of the Gaussian family with an identity-link function. All outcome variables were normally distributed (skewness < 0.2). For binary outcomes (stunting, wasting, and underweight) we used GEE models with the Poisson distribution family with a log-link function to calculate the risk ratio (RR) [34]. For all models, we specified a robust estimator of variance and an unstructured correlation structure. The exposure variable was tCys concentration; potential confounding variables were age (in months), breastfeeding (yes/no), baseline plasma folate concentration, years of education of mother and father, household income (in US$), family size, intervention group, and a composite measure for vitamin B_12_ status (3cB12). In addition, we assessed an interaction between breastfeeding status and age. All potential confounders were assessed in bivariable models, and all variables significant at a level of 0.2 were taken into a multivariable model simultaneously. All variables not significant at a level of 0.05 were taken out one by one and removed if the coefficient for tCys did not change considerably (>15%). After all non-significant variables were removed from the multivariable model, all variables initially not taken into the model, were entered one by one again and retained if now significant (*p* < 0.05).

We used generalized additive models (GAM) to depict a potential non-linear relationship between tCys concentration and anthropometric indices (HAZ, WHZ or WAZ). We adjusted for all confounders identified in the GEE model (of the Gaussian family) and deleted values <2.5th percentile and ≥97.5th percentile for tCys to avoid overfitting at the extremes.

## 3. Results

### 3.1. Baseline Characteristics

Plasma total cysteine concentrations and anthropometric measurements at both time points were available for 2102 children (Figure 1). The demographic, nutritional, and socio-economic characteristics of the included children are summarized in Table 1. On average, children were 15 months old at enrolment, and the majority were breastfed (69%). The families had on average five family members, but family size varied substantially (between 2 and 19 persons living together). Fathers had on average more years of schooling (median 8 years) than mothers (median 5 years). A high proportion of the children were defined as malnourished at enrolment; 41% were stunted (HAZ ≤ −2), 19% were wasted (WHZ ≤ −2) and 42% underweight (WAZ ≤ −2). At the end of the study (4 months after enrollment), the prevalence was 46%, 20% and 43%, respectively. A display of the distribution of all anthropometric indicators can be found in Appendix A. All characteristics shown in Table 1 differed between those children that had a plasma cysteine concentration ≤25th percentile compared to >25 percentile, except family size, sex, and vitamin B_12_ concentration.

### 3.2. Cysteine and Undernutrition

Plasma total cysteine (tCys) at baseline was significantly associated with HAZ, WHZ, and WAZ after adjusting for relevant confounders (*p* < 0.001). With one unit increase in cysteine concentration (in µmol/L), the Z-scores for all anthropometric indicators increased by 0.003–0.005 Z-scores (Table 2; GEE 1). The GAM plots also confirm this general trend of an increase in anthropometric status with increasing tCys concentration (Figure 2). Having a low tCys concentrations (≤25th percentile) was associated with a lower Z-score for HAZ (−0.28; 95% Confidence Interval (CI): −0.39, −0.18), for WHZ (−0.10; 95% CI: −0.19, −0.01), and for WAZ (−0.21; 95% CI: −0.31, −0.11), also after adjustment for relevant confounders (Table 2; GEE 2). In those with a low cysteine concentration (<25th percentile), the risk was significantly higher for being stunted (Relative Risk (RR) 1.22, 95% CI: 1.12, 1.33) and underweight (RR 1.21, 95% CI: 1.10, 1.33), respectively (Table 2; GEE 3). The risk of being wasted was higher in those children with a low tCys, but it did not reach statistical significance when adjusted for relevant confounders (RR 1.12, 95% CI: 0.95, 1.32).

Information for this analysis was available for 85% of all the children enrolled in the original trial. In children with no available information on anthropometry (*n* = 256), tCys concentration did not differ from children with this information. Those children without information on tCys (*n* = 216), had slightly higher mean Z-scores for all anthropometric indicators at baseline (HAZ, WHZ and WAZ). Density plots can be seen in the Appendix A. All other characteristics included in Table 1 did not differ significantly, nor did the means of the anthropometric indicators at endpoint of the study.

## 4. Discussion

In our study sample of 2102 children aged 6–30 months living in low-to-middle socioeconomic neighborhoods of New Delhi, India, plasma total cysteine (tCys) concentration was positively associated with height-for-age (HAZ), weight-for-height (WHZ) and weight-for-age (WAZ). Children with low tCys concentrations (≤25th percentile) had a ~20% higher risk of being stunted, or underweight. The risk of wasting did not reach statistical significance.

The low intake of high-quality protein intake, as well as amino acid, is associated with poor childhood growth [9,10,11,12,13,14]. In a targeted metabolomics approach, Semba and colleagues [11] showed that 16 of 19 measured proteinogenic amino acids were significantly associated with stunting in Malawian children. They suggested a potential biological framework through the mechanistic target of rapamycin complex C1 (mTORC1) which has been shown to repress the synthesis of proteins, lipids and thus limited cell growth in the absence of amino acids [14]. However, cysteine was not included in their study and we are not aware of any other studies on linear growth in children in relation to cysteine. In mice experiments, cysteine administration resulted in increased growth plate thickness, probably through an upregulation of IGF-1 [16], which makes a direct effect of cysteine on linear growth plausible. While the regression coefficients in our analysis are significant, the effect size of GEE 1 using continuous variables can seem small; however, it needs to be emphasized that this is not surprising for a single nutrient. It was estimated that, even if all nutrition-specific interventions that have a solid scientific evidence base would be implemented in 90% of all children in the world, childhood stunting would only be reduced by 20% [35]. Indeed, many interventions have had a modest or no effect on child growth [36]. If cysteine would improve linear growth in children as shown in the animal experiment [16], this could be a small contribution to alleviating the prevailing burden of stunting in many countries such as India. However, further studies are needed to confirm our findings, addressing our limitations, and to subsequently establish if a randomized supplementation would show any effect.

The role of cysteine in kwashiorkor, a form of severe malnutrition, has been the focus in some studies [21,22,23] set within the framework of the antioxidant theory proposed by Golden et al. [37]. These suggest that severely malnourished children require more cysteine as a precursor of glutathione (GSH) for anti-oxidant defense, as a building block for acute-phase proteins or mucins for an adequate immune response, or for gut mucosal proteins supporting to reverse the impaired gut function associated with severe acute malnutrition [21]. However, the only randomized controlled trial identified through a Cochrane Review [38] did not show any effects of supplementary antioxidant micronutrients including cysteine to Malawian children on the occurrence of kwashiorkor [39].

Other relevant literature on cysteine comes from studies on growth restrictions in utero. In animal studies, supplementing pregnant piglets with N-acetylcysteine (NAC), a precursor converted to cysteine in the liver, prevented intra-uterine growth restriction (IUGR) [40]. The administration to neonatal piglets with IUGR reduced mucosa damage. The authors suggest protection of the intestinal morphology, through a reduced mitochondrial swelling and prevention of reactive oxygen species (ROS) overproduction. Further, they suggest improvements in energy metabolism and antioxidant effects on the intestine [41]. In studies involving humans, maternal cysteine levels have been associated with low birth weight. One study including pregnant women found an association of low maternal cysteine (and GSH) concentrations with pre-term birth as well as cysteine concentration in the offspring [25]. However, other studies reported an association between high levels of maternal plasma tCys and pre-eclampsia, pre-term birth, low birth weight, and length at birth [24,42]. We did not have any data on pregnancy outcomes and can therefore not make any statement about the timing of the onset of the poor anthropometric status.

The GAM plots in our work (Figure 2) show a general trend of an increase in anthropometric Z-score (centered around the mean) with increasing tCys concentration. Although the lines are not completely linear, they support the notion of an approximately linear relationship previously reported in adults [43]. However, this is in contrast to a study in children aged 4–19 years, where total cysteine was significantly associated with body fat only in overweight/obese children but not in those with normal weight [44]. There were very few children in the upper range of weight-for-height in our sample (7 children had a WHZ >2 at baseline).

### Limitations and Strength

Data for the study come from a randomized controlled trial on zinc supplementation which limits the generalizability. Information for this analysis was available for 85% of all the children enrolled in the original trial. Although children without information on tCys had slightly higher mean Z-scores for all anthropometric indicators, all other characteristics in our analysis did not differ significantly; thus, we do not expect that missing data on tCys would bias our results considerably. The strength of our study was that we could base our analysis of high-quality data from a community-based cohort. This investigation was designed as an observational study, and although we considered several relevant confounders, we cannot exclude residual confounding. Half of the children were supplemented with zinc in this study. Although in the main trial the zinc group had a significantly reduced incidence of diarrhea and respiratory tract infections compared to those receiving placebo [26,45], oral zinc did not result in improved growth [46]. When included in our models, zinc supplementation was not significantly associated with either of the anthropometric indicators, and therefore not included in the final multivariable models.

The 10% variation in the cysteine assay is a potential limitation; however, this non-differential misclassification in the exposure decreased the precision of our effect measure estimates and could attenuate the observed associations between tCys and growth [47]. We used the 25th percentile as a cut-off to indicate a low concentration of tCys. There are no recognized cut-offs for cysteine to indicate a deficiency and therefore we based the choice on a statistical approach to capture the part of a population with increased vulnerability. Although the GAM plots do not show any clear cut-point at which the risk for poor anthropometric status increases, a slightly steeper curve at the left tail until around the 25th percentile (i.e., 164 μmol/L) supports our decision.

The low intake of high-quality protein intake, as well as amino acid, is associated with poor childhood growth [9,10,11,12,13,14]. We did not include dietary intake in this analysis, and thus, we cannot exclude that tCys concentrations were only a marker of the overall quality of the diet and the role of other dietary components remains unclear. Moon et al. [16] hypothesize that part of the effect of cysteine supplementation on bone growth in mice was due to sulfur. The intake of methionine, an essential sulfur amino acid, can affect cysteine concentrations, as it is a precursor thereof. Cysteine can replace parts of methionine intake, but not all of it [48]. We did not have data on methionine concentrations, but we account for its potential association through adjusting for tHcy and cobalamin in our multivariate models, both of which are necessary for the transsulfuration pathway from methionine to cysteine. The main source of dietary cysteine is animal source proteins [48] and diets in this population are commonly characterized by low intake of animal products [49]. In our study, the mean (SD) tCys concentration was 179 (25) μmol/L. There is no recognized normal range for cysteine and we did not find any directly comparable study reporting on serum tCys concentrations. For adults, Stipanuk [48] states that plasma tCys concentration in healthy adults would range from about 220–320 μmol/L. In a hospital-based study among 48 pre-term neonates with gastrointestinal disease, the mean serum tCys concentration was 153 (25), and among 288 US children who were on average 10 years old, the mean concentration was 193 μmol/L [44]. Cysteine dioxygenase (CDO), an enzyme involved in cysteine catabolism, is one of the most highly regulated enzymes responding to changes in dietary intake within hours, suggesting that cysteine homeostasis plays a central role in the human body [50]. However, Jones, et al. [51] state that insufficient intake of sulfur amino acid (such as cysteine) over a long time cannot be masked by recent intakes.

We did not include morbidity data in our analyses and cannot assess if cysteine concentration is associated with ponderal and linear growth via inflammation, mucosal damage, or other morbidities. A large part of the pediatric populations in low- and middle-income countries are affected by environmental enteric dysfunction (EED), characterized by structural and functional changes in the small intestines, leading to chronic inflammation and malabsorption of nutrients. EED is increasingly recognized as a risk factor for malnutrition [52,53,54,55]. The association of cysteine with reduced oxidative stress, inflammation and cell apoptosis, as well as improved integrity of the gut mucosa [56,57] and the potential role thereof in the complex etiology of malnutrition, warrants further research on metabolic alterations involving cysteine.

## 5. Conclusions

In our community-based population, young Indian children with a low plasma total cysteine concentration had an elevated risk of a poor anthropometric status. Based on our results, we cannot conclude that cysteine deficiency causes poor growth. Whether or not the association between cysteine and growth is due to metabolic processes related to nutritional status or due to residual confounding is not known. The role of cysteine in malnutrition should be estimated in more comprehensive studies, including several biomarkers, dietary intake, and preferably in populations representing the whole range of nutritional status during various critical life-cycle windows.

## Figures and Tables

**Figure 1 nutrients-12-03146-f001:**
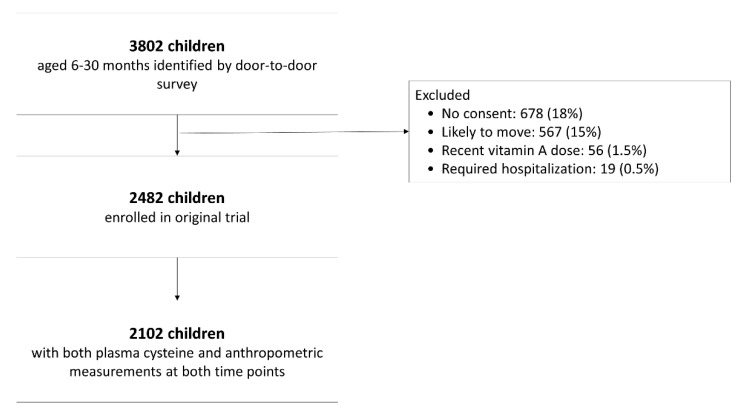
Study participant flow chart.

**Figure 2 nutrients-12-03146-f002:**
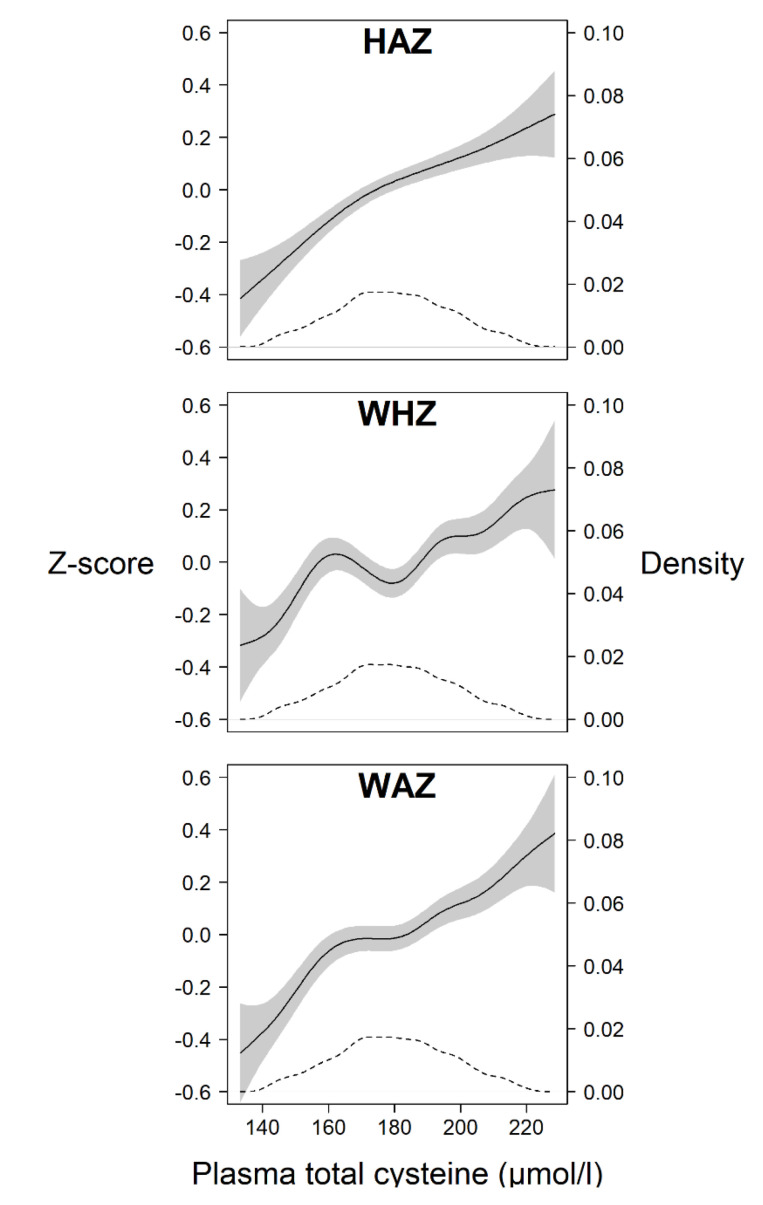
Generalized additive model (GAM) plots showing the relation between plasma total cysteine (in µmol/L) and Z-scores centred around the median (left *y*-axis) for the anthropometric indicators height-for-age (HAZ), weight-for-height (WHZ) and weight-for-age (WAZ) (solid lines), in children aged 6–30 months living in Delhi, India. All anthropometric indicators are calculated based on the WHO Child Growth Standards. The grey area depicts the 95% confidence interval and the dashed line depicts the density distribution of the plasma total cysteine values (right *y*-axis).

**Table 1 nutrients-12-03146-t001:** Baseline characteristics of 2102 children aged 6–30 months living in New Delhi, India included in the analysis.

Characteristic	All Children (*n* = 2102)	<25th Percentile Cysteine (*n* = 1576)	≥25th Percentile Cysteine (*n* = 526)
Mean age (SD), months	15.4 (7.5)	17.0 (7.1) **	14.8 (7.6)
Proportion male, %	52.1	49.6	53.0
Proportion breastfed, %	69.1	53.4 **	74.3
Mean years of schooling of mother (SD)	5.2 (4.5)	4.8 (4.5) *	5.3 (4.6)
Mean years of schooling of father (SD)	8.4 (4.1)	7.8 (4.2) **	8.6 (4.0)
Median family size (IQR)	5 (4–7)	6 (4–7)	5 (4–7)
Median annual household income (IQR), US$	720 (480–1080)	660 (480–960) *	720 (480–1200)
Mean HAZ at enrolment (SD)	−1.78 (1.18)	−2.10 (1.19) **	−1.67 (1.16)
Mean HAZ at study end (SD)	−1.92 (1.14)	−2.24 (1.13) **	−1.81 (1.12)
Mean WHZ at enrolment (SD)	−1.15 (1.03)	−1.35 (0.98) **	−1.09 (1.03)
Mean WHZ at study end (SD)	−1.19 (0.95)	−1.21 (0.94)	−1.18 (0.96)
Mean WAZ at enrolment (SD)	−1.81 (1.09)	−2.11 (1.07) **	−1.71 (1.08)
Mean WAZ at study end (SD)	−1.86 (1.04)	−2.06 (1.05) **	−1.80 (1.04)
Mean plasma total cysteine concentration (SD), μmol/L	179 (25)	149 (14)	190 (19)
Median plasma cobalamin concentration (IQR), pmol/L	206 (141–300)	213 (145–308)	204 (140–297)
Median plasma folate concentration (IQR), nmol/L	10.6 (6.4–19.8)	7.3 (4.8–12.1) **	12.0 (7.3–22.7)
Median plasma MMA concentration (IQR), μmol/L	0.65 (0.37–1.29)	0.55 (0.33–1.13) *	0.67 (0.38–1.34)
Median plasma tHcy concentration (IQR), μmol/L	10.8 (8.3–14.8)	9.6 (7.6–13.1) **	11.2 (8.6–15.5)
Mean 3cB12 (SD)	−1.03 (1.01)	−0.84 (1.06)	−1.09 (0.99)

* Statistically significant difference to children in ≥25th percentile category, *p*-value < 0.05; ** Statistically significant difference to children in ≥25th percentile category, *p*-value < 0.001; Abbreviations: HAZ = height-for-age Z-score, IQR = interquartile range, MMA = methylmalonic acid, SD = standard deviation, tHcy = total homocysteine, WAZ = weight-for-age Z-score, WHZ = weight-for-height Z-score, 3cB12 = combined indicator for vitamin B_12_.

**Table 2 nutrients-12-03146-t002:** Association between plasma total cysteine and selected repeated anthropometric indicators among 2102 children aged 6–30 months living in Dakshinpuri, New Delhi, India estimated in generalized estimating equations (GEE) with robust variance estimation.

	GEE 1 ^1^	GEE 2 ^2^	GEE 3 ^3^
Outcome	Coeff (95% CI)	Coeff (95% CI)	RR (95% CI)
**HAZ**			
Unadjusted	0.0083 (0.0063, 0.0102)	−0.42 (−0.53, −0.31)	1.37 (1.26, 1.50)
Adjusted ^4^	0.0053 (0.0034, 0.0071)	−0.28 (−0.39, −0.18)	1.22 (1.12, 1.33)
**WHZ**			
Unadjusted	0.0046 (0.0031, 0.0061)	−0.16 (−0.25, −0.08)	1.17 (1.01, 1.38)
Adjusted ^5^	0.0034 (0.0018, 0.0049)	−0.10 (−0.19, −0.01)	1.12 (0.95, 1.32)
**WAZ**			
Unadjusted	0.0073 (0.0056, 0.0090)	−0.33 (−0.43, −0.23)	1.32 (1.20, 1.44)
Adjusted ^6^	0.0050 (0.0033, 0.0066)	−0.21 (−0.31, −0 11)	1.21 (1.10, 1.33)

^1^ In this GEE model, the outcome is entered as continuous Z-score (Gaussian distribution with identity link); plasma total cysteine is entered as continuous variable (mmol/L). ^2^ In this GEE model, the outcome is entered as continuous Z-score (Gaussian distribution with identity link); plasma total cysteine is entered as dichotomous variable (≤25th percentile compared to >25th percentile). ^3^ In this GEE model, the outcome is entered as dichotomous variable (≤−2 Z-scores compared to >−2 Z-scores) specified as Poisson distribution with a log link); plasma total cysteine is entered as dichotomous variable (≤25th percentile compared to >25th percentile). ^4^ adjusted for age, breastfeeding status, interaction between age and breastfeeding, mother’s education, father’s education, number of family members, and B_12_ at baseline. ^5^ adjusted for age, breastfeeding status, interaction between age and breastfeeding, mother’s education, number of family members, folate, and B_12_ at baseline. ^6^ adjusted for age, breastfeeding status, interaction between age and breastfeeding, mother’s education, father’s education, number of family members, folate, and B_12_ at baseline; Abbreviations: CI: confidence interval; coeff: regression coefficient; GEE: generalized estimating equations; HAZ: height-for-age Z-score; RR: relative risk; WAZ: weight-for-age Z-score; WHZ: weight-for-height Z-score.

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
