# Peer review of "Association of Plasma Total Cysteine and Anthropometric Status in 6–30 Months Old Indian Children"

_nutrients, 2020, doi:10.3390/nu12103146_

Round 1
Reviewer 1 Report
In general, this paper is compact and clearly written. It is of value that the authors analyse existing data for relevant purposes. No doubt that malnutrition is an important and relevant issue to deal with at the site of the study.
However, although some mechanisms of cysteine metabolism and relation with anthropometry have been described, the relevance of this study remains unclear. More in depth background information would be necessary to underpin the relevance of this topic.
The following items shoud be addressed:
- In line 54 it is stated that “In children, the association between cysteine and linear growth remains to be elucidated.” Please clarify why in the view of the authors this is relevant to know and what would it help for daily practice? In particular, what does any outcome of this study change in treating malnutrition, or in particular, in India?
- It is not clear to what extent plasma levels of cysteine reflect (nutritional) status of this amino acid on whole body level and why this amino acid in particular is the focus of the study. In addition, levels of its precursor methionine are not taken into account to make the picture complete.
- It is not clear how was checked for normality of the data (section 2.3).
- Growth data of two time points, i.e. enrollment and at 4 months, were used for the analyses and it was stated that ‘anthropometric indices’ were used as ‘time-varying outcomes’ (lines 127-128), whereas only tCys concentrations from enrollment was used. However, from the description of the models it does not become clear how has been dealt with the changes in anthropometric values over the 4-mo time span. Also, it has not made clear how multiple measures in the same child has been taken into account in the analyses. This should be described clearly in the methods.
- A risk ratio has been calculated for the relation between plasma cystein levels and the three anthropometric indices (line 133). However, the determinant used in this analysis (plasma cysteine) was not a determinant of the main study (RCT with zinc intervention versus placebo). In addition, calculating a relative risk implies some sort of relationship over time, which in this study is not clearly described. One would expect that for example changes in anthropometric indices have been taken into account, however, no such information is available in the paper.
- In order to increase transperancy of the data of this study, additional data should be clearly presented either in Table 1 (preferably) or in the text on: 4 month data of anthropometric indices, absolute data of weight, height/length, -2 z scores of the anthropometric indices, mean plasma cysteine concentrations for each group separately (i.e. <25th percentile and above), composite score of cobalamine (3cB12). Actually, to increase transparancy further, scatter plots presenting the distribution of the anthropometric measures would be valuable.
- How has the large difference between the number of children within the <25th percentile cysteine versus >25th percentile been taken into account? As can be observed from table 2 the coefficients from model 1 and 2 are, although statistically significant, very small.
- Do the authors have anticipated on the effect of the 10% variation in the cysteine assay in the overall regression analyses?
- Minor remark (Figure 2): what do the numbers at the right site of the graphs represent?
- It was concluded (line 241) that children with low cysteine concentrations have a higher risk of being wasted, although not significant. Besides that the RR analysis is questionable, stating outcomes for being higher while not statistically significant should be avoided.
Moreover, the conclusion of an elevated risk (line 330) is questionable and should be rephrased. - Could the authors elaborate on determining plasma levels such as cysteine and cysteine status, in order to state the relevance of this study. In addition, what is the relevance of choosing the 25th percentile, i.e. 164 micromol/L, for cut-off? Comparing with data in literature, could the authors elaborate on how high/low this level is? Why not use the continuous data of model 1?
- In addition, if model 1 shows that one unit increase in cysteine concentrations increased anthropometric Z-scores by 0.003-0.005, what do this imply in terms of wasting, stunting or malnutrition?
- Causality could not be determined based on this study, so please rephrase stating that “…and cannot confirm if cysteine concentration might have affected ….” (line 318-319).
- The relevance of performing “More comprehensive studies, including several biomarkers, dietary intake, and preferably in populations representing the whole range of nutritional status during various critical life-cycle windows, are needed to extend our understanding of the complex mechanisms related to malnutrition.”(line 332-335) is not made sufficiently clear and an overall view of cysteine levels in relation to the malnutrition phenomenon is neclected.

Reviewer 2 Report
In a prospective cohort study, the authors have examined the association between plasma total cysteine and anthropometric status in Indian children. Their data show an elevated risk of malnutrition in children with a low plasma total cysteine concentration.
The study is done well and the data are clearly presented.
The authors are well aware of the study limitations and state them fairly in the discussion
- The manuscript requires moderate rewrite for language.
- The authors should also discuss more on the link between B12, folates and cysteine both in the introduction and in the discussion sections.
